# Semiconductor photocatalysis to engineering deuterated *N*-alkyl pharmaceuticals enabled by synergistic activation of water and alkanols

Zhaofei Zhang[1,5], Chuntian Qiu[1,5], Yangsen Xu [1], Qing Han[2,3], Junwang Tang[2], Kian Ping Loh [4] & Chenliang Su [1✉]

Precisely controlled deuterium labeling at specific sites of *N*-alkyl drugs is crucial in drug-development as over 50% of the top-selling drugs contain *N*-alkyl groups, in which it is very challenging to selectively replace protons with deuterium atoms. With the goal of achieving controllable isotope-labeling in *N*-alkylated amines, we herein rationally design photocatalytic water-splitting to furnish [H] or [D] and isotope alkanol-oxidation by photoexcited electron-hole pairs on a polymeric semiconductor. The controlled installation of *N*-$CH_3$, -$CDH_2$, -$CD_2H$, -$CD_3$, and -$^{13}CH_3$ groups into pharmaceutical amines thus has been demonstrated by tuning isotopic water and methanol. More than 50 examples with a wide range of functionalities are presented, demonstrating the universal applicability and mildness of this strategy. Gram-scale production has been realized, paving the way for the practical photosynthesis of pharmaceuticals.

[1] International Collaborative Laboratory of 2D Materials for Optoelectronic Science & Technology of Ministry of Education, Engineering Technology Research Center for 2D Materials Information Functional Devices and Systems of Guangdong Province, Institute of Microscale Optoeletronics, Shenzhen University, 518060 Shenzhen, China. [2] Department of Chemical Engineering, University College London, Torrington Place, London WC1E 7JE, UK. [3] Key Laboratory of Photoelectronic/Electrophotonic, Conversion Materials, Key Laboratory of Cluster Science, Ministry of Education, School of Chemistry, Beijing Institute of Technology, 100081 Beijing, China. [4] Department of Chemistry and Centre for Advanced 2D Materials (CA2DM), National University of Singapore, 3 Science Drive 3, Singapore 117543, Singapore. [5] These authors contributed equally: Zhaofei Zhang, Chuntian Qiu. ✉email: chmsuc@szu.edu.cn

sotope labeling plays vital roles in various fields in synthetic chemistry, quantitative LC–MS/MS analysis, and the life sciences[1–6]. The higher stability of C–D bonds than C–H bonds because of the deuterium kinetic isotope effect (DKIE) motivates the need for a "deuterium switch" in drug synthesis to improve biological properties, such as pharmacokinetics, pharmacodynamics (PK/PD), and metabolic stability[7–11]. In 2017, the first deuterium-labeled drug, deutetrabenazine, was approved by the FDA and initiated a new era of deuterated clinical drug development[12]. Among the myriad of commercial drugs, over 50% of the top sellers contain $N$-alkyl amine units[13], and the $N$-dealkylation metabolized cytochrome P450 (CYP450) are commonly found in such $N$-alkyl drugs and other bioactive molecules[14–19]. Thus, deuterium substitution of $N$-alkyl groups in $N$-alkyl drug molecules could contribute to slow down the N–C bond cleavage, and impacts their pharmacodynamic properties and improve pesticide effects[20–24]. In this regard, the precision synthesis of

drug analogs with deuterated $N$-alkyl amine units holds great promise and has been attracting increasing interest (Fig. 1a and Supplementary Fig. 1).

Traditional approaches to $N$-alkyl drugs usually require the use of deuterated alkylation reagents such as $CD_3I$ (Supplementary Fig. 2a). The substitution is of interest as these alkylation reagents are highly toxic, carcinogenic, and volatile[25–28], which generally cause high costs and waste production. In addition, these reactions often suffered from excess methylation leading to ammonium salts[29,30]. Reduction of $N$-$CO_2R$ moieties with $LiAlD_4$ is another effective approach that has good potential for the introduction of $N$-$CD_3$ group without formation of ammonium salts[20]. However, introduction of extra functional group, use of hazard and strong reducing reagent, and poor functionality tolerance limit its practical application (Supplementary Fig. 2a). Recently, catalytic hydrogen isotope exchange (HIE) of $\alpha$- or $\beta$-amines[31–37] has been emerged as a promising way to incorporate

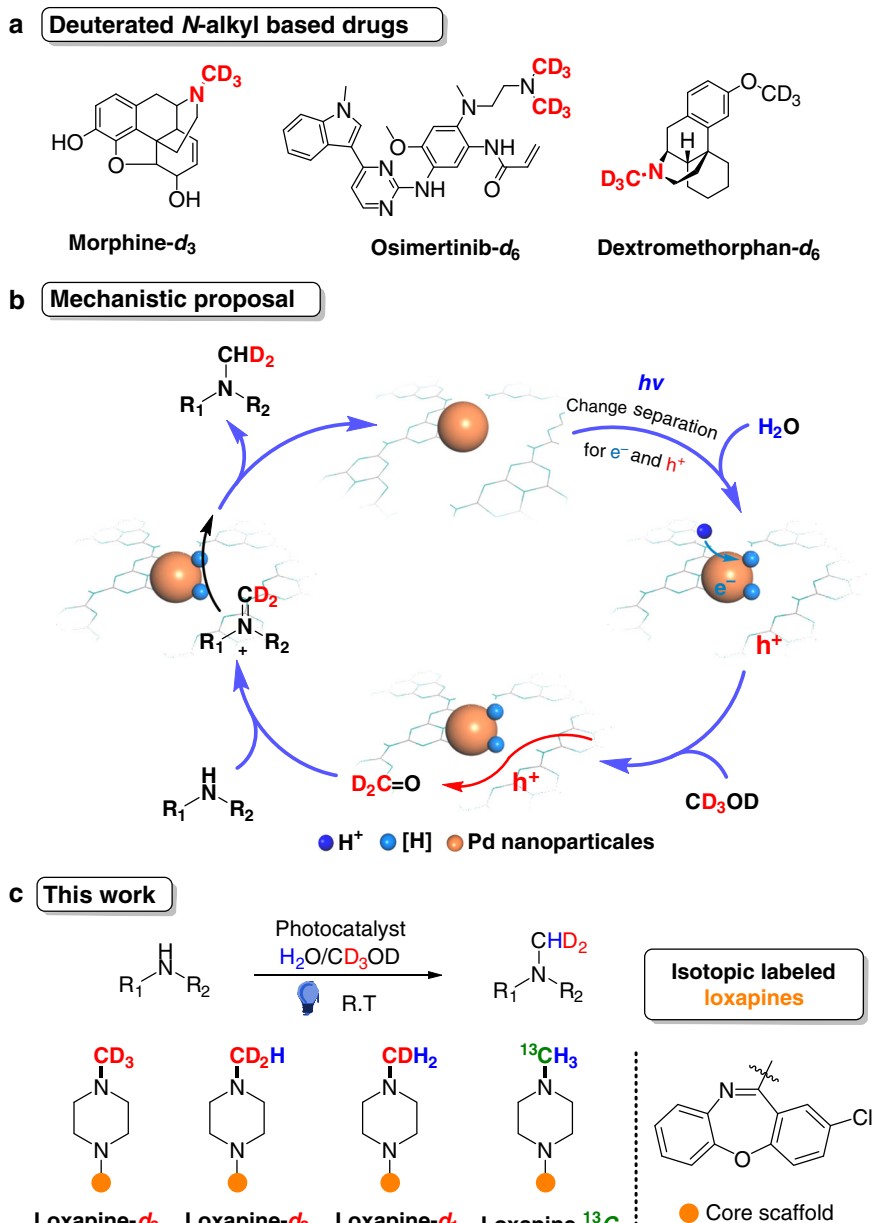

**Fig. 1 Synthesis of deuterated $N$-alkylation drugs based on photocatalytic water-splitting process. a** typical deuterated $N$-alkyl based drugs. **b** Mechanistic proposal of the controllable isotope-labeled $N$-methylation of amines by the synergistic utilization of electrons and holes on a semiconductor photocatalyst. **c** This work, and the example of practical synthesis of isotope-labeled loxapines.

multi-deuterium or tritium atoms into *N*-alkyl amine-based drugs (Supplementary Fig. 2b). For example, MacMillan group[38] reported a powerful photo-redox mediated HIE reaction which could efficiently and selectively install deuterium or tritium at α-amino sp[3] C–H bonds of the *N*-alkyl amine-based drug molecules. In this protocol, the *α*-position of amines is oxidized by a molecular photocatalyst to yield α-amino radical, which was then trapped by the hydrogen atom transfer (HAT) catalysis mediated the abstraction of deuterium from $D_2O$ or $T_2O$ to furnish α-deuterated or tritiated amine product. Multi-deuterium atoms incorporation at all *α*-position of pharmaceutical amines (more than 4.0 deuteriums per molecule) with a wide range of D-incorporation ratio (from 1 to 91%) is generally occurred. Still, the development of a general and mild method for the substitution of the traditional deuterated alkylation from toxic deuterated reagents like $CD_3I$ is in high demand to effectively and selectively functionalize pharmaceutical amines[39–43]. Further, the precise control of deuterium atoms number at the *α*-position of *N*-alkyl drugs with high deuterium incorporation currently remain unexplored, while it is particularly attractive for their potential use in mechanistic and metabolic studies[44,45].

Semiconductor photocatalysts, which provide redox center on the surface upon light irradiation can be designed to decompose $H_2O/D_2O$ to furnish reductive [H]/[D] and simultaneously oxidize organics by the photoexited electron–hole pairs[46–49]. Synergistic utilization of those reductive [H]/[D] and reactive organic species holds great potential for production of deuterated chemicals and pharmaceuticals, e.g. D-labeling *N*-alkyl pharmaceuticals, from isotopic water and organics. Polymeric carbon nitride (PCN) is a nontoxic, highly stable, low-cost, and scalable polymer semiconductor with a suitable redox window [from approximately +1.2 V to −1.5 V vs. saturated calomel electrode (SCE)][50,51]. These characteristics define PCN as an ideal semiconductor photoredox catalyst for effective water splitting coupled with controlled oxidation by photoexcited electron–hole pairs. Herein, we utilize highly crystalline PCN as a semiconductor photocatalyst for the sustainable synthesis of *N*-alkyl chemicals and drugs with well-controlled isotope labeling[52]. Upon visible-light irradiation, electron–hole pairs are generated on crystalline PCN. Photogenerated electrons are transferred to the anchored Pd nanoparticles and utilized to reduce water to furnish absorbed [H]/[D] species. Meanwhile, photogenerated holes with appropriate oxidative ability are designed to selectively oxidize isotopic alkanols, furnishing isotopic aldehydes for aldehyde-amine condensation to produce imine intermediates. These imines are subsequently reduced by [H]/[D] from water splitting, producing corresponding *N*-alkyl chemicals and drugs (Fig. 1b). Compared to traditional approaches from deuterated

alkylation reagents, this photocatalytic strategy exhibits several advantages: (a) the low-cost and sustainable isotopic water and alkanol is proposed as a combined deuterated alkylation reagent, (b) benefiting from this unique design, precise controlling the number of deuterium atoms (i.e., *N*-$CD_3$, $CD_2H$ and $CDH_2$) at the metabolic position of *N*-Me drugs is enabled by simply tuning the isotopic water and methanol (Fig. 1c); (c) excess deuterated methylation leading to ammonium salts could be effectively avoided; (d) finally, this heterogeneous process exhibits high yields, broad reaction scope, excellent one-step D-incorporation, and scalable production, thus paving the way towards deuterated drug studies and developments.

## Results

**Controllable installation of *N*-CD₃ groups of *p*-toluidines and diphenylamines.** We started our investigation by screening conditions for the water-splitting-based *N*-methylation of amines using highly crystalline PCN (CPCN) as the semiconductor photocatalyst[53], water and methanol as the green methylating reagents, and *p*-toluidine as the amine. The optimized conditions are summarized in Supplementary Table 1, where the *N*-methylation product of *p*-toluidine, *N*,*N*-$(CH_3)_2$ *p*-toluidine, was obtained in 94% yield. Using the optimized conditions, isotopic water and methanol were used to investigate the synthesis of deuterated compounds and the reaction pathway. Generally, multiple reaction processes are required to achieve high deuteration content in the production of deuterated chemicals and pharmaceuticals. Here, the use of $D_2O$ and $CD_3OD$ afforded *N*,*N*-$(CD_3)_2$ *p*-toluidine in 89% yield with high D incorporation (97%). To trace the deuterium source, $H_2O/CD_3OD$ was used, which afforded *N*,*N*-$(CD_2H)_2$-*p*-toluidine in 91% yield, with nearly quantitative D-incorporation (>99%). The obtained partially deuterium-labeled product suggests that $CD_3OH/CD_3OD$ are probably oxidized to $[D_2C=O]$ by photogenerated holes, which is consistent with the mechanism of photocatalytic water splitting using methanol as the sacrificial agent[49,50]. Aldehyde-amine condensation of $[D_2C=O]$ and *p*-toluidine occurs to furnish imine intermediates for sequential hydrogenation by reductive [H] from water splitting (Fig. 1c). The secondary amine intermediate then undergoes another aldehyde-amine condensation followed by hydrogenation with [H], producing the corresponding *N*,*N*-$(CD_2H)_2$-*p*-toluidine product. Consistent with the aforementioned reaction pathway, using the $D_2O/CH_3OD$ system could introduce *N*-$CDH_2$ groups (91%) with high D content (>99%). The controllable D-labeled *N*-alkylation of secondary amines was also examined, affording *N*-$CH_3$, -$CD_3$, -$CD_2H$, and -$CDH_2$ diphenylamines in high yields (74–94% yields) with excellent D incorporation (>97%) (Fig. 2). Our results show

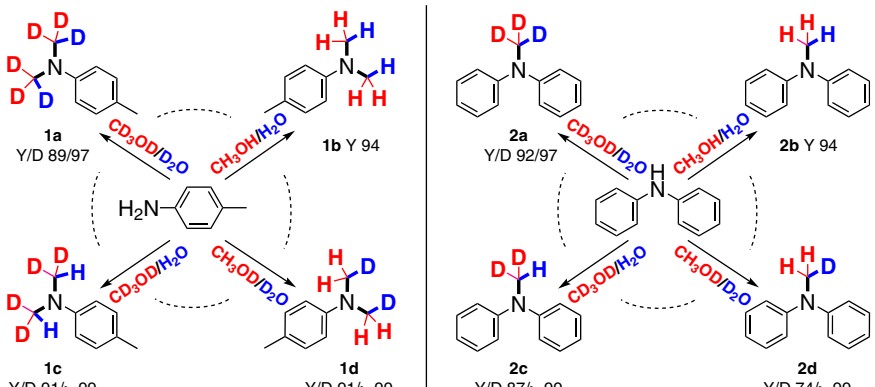

**Fig. 2 Controllable D-labeled *N*-methylation of primary and secondary amines.** Y refers to isolated yields of deuterated products. D refers to D-incorporation percentages based on the calculation of ¹HNMR.

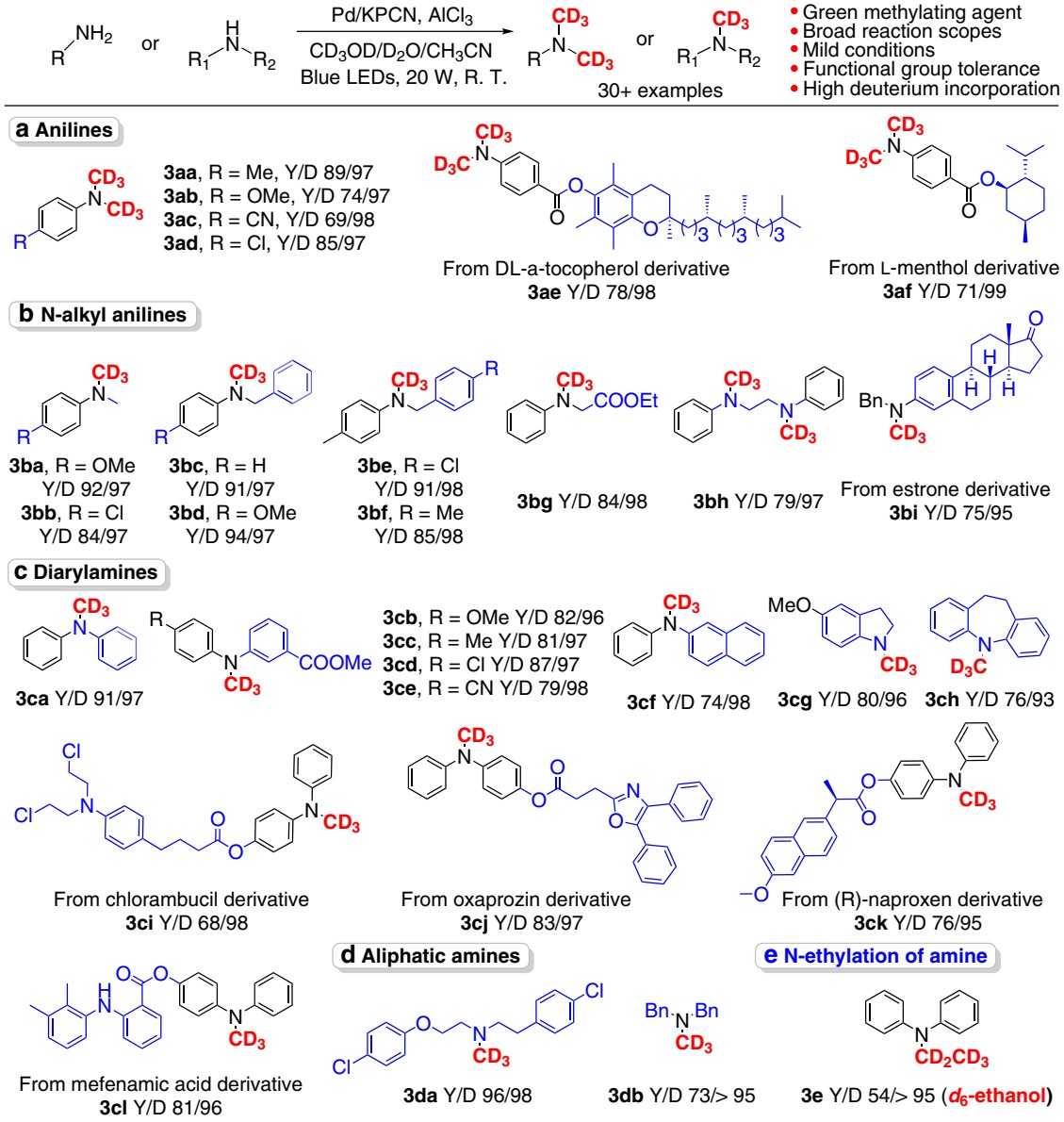

**Fig. 3 Substrate scopes of the *N*-trideuteromethylation of amines.** Y represents yield. D represents D-incorporation percentage. Reaction conditions: 0.4 mmol amine, 25 mg of Pd/CPCN, 0.3 mmol AlCl₃, Acetonitrile/D₂O/CD₃OD = 2 ml/1.5 ml/1.0 ml, Blue LEDs, 20 W, rt. **a** Synthesis of deuterated annilines. b Synthesis of deuterated *N*-alkyl annilines. **c** Synthesis of deuterated diarylamines. d Synthesis of aliphatic amines. **e** *N*-ethylation of diarylamine.

convincingly that the number of deuterium atoms installed at the *N*-methyl groups can be precisely controlled, thus showing great promise for the precise introduction of deuterium atoms in the specific position of *N*-Me-based drugs.

**Photocatalytic water-splitting-based *N*-methylation of amines.** Next, the generality of the water-splitting-based *N*-trideuteromethylation of amines was tested by synthesizing valuable *N*-CD₃-based deuterated chemicals and pharmaceutical derivatives (Fig. 3). Primary amines underwent two *N*-trideuteromethylation reactions, providing products with *N,N*-(CD₃)₂ units with excellent D incorporation (97-99%) (Fig. 3, **3aa-3af**). The use of aniline substrates bearing both electron-donating groups (*p*-Me, *p*-OMe) and electron-withdrawing groups (*p*-CN, *p*-Cl) produced the corresponding *N,N*-(CD₃)₂-anilines in 69–89% yields (**3aa-3ad**). Sensitive substrates with alkyl chiral centers (**3ae** and **3af**) were compatible and unperturbed. Since most *N*-alkyl drugs are

fabricated from secondary amines via *N*-alkylation reactions, the *N*-trideuteromethylation of secondary amines was investigated with great interest. To our delight, this protocol with secondary amines exhibits a broad reaction scope, good functional group tolerance and excellent D incorporation. *N*-alkyl anilines, including substituted *N*-Me anilines and *N*-Bn anilines, furnished the corresponding products with high D incorporation (91–98%) and in excellent yields (84–94%). *N*-trideuteromethylation of ethyl phenylglycinate (**3bg**), a representative amino acid derivative, as well as estrone derivate (**3bi**) was achieved, attesting to the ability to deuterate bioactive molecules. For the diamine substrate **3bh**, *di*-CD₃ was simultaneously introduced in 71% yield. This protocol was also applicable to a wide range of diary amines bearing substituted phenyl (**3ca-3ce**), naphthyl (**3cf**), and pharmaceutical units such as chlorambucil (**3ci**), oxaprozin (**3cj**), and (R)-naproxen (**3ck**). A steric effect-controlled highly chemoselective *N*-trideuteromethylation of diary amines is observed (**3cl**). The *N*-CD₃ incorporation of heterocyclic amines such as indoline

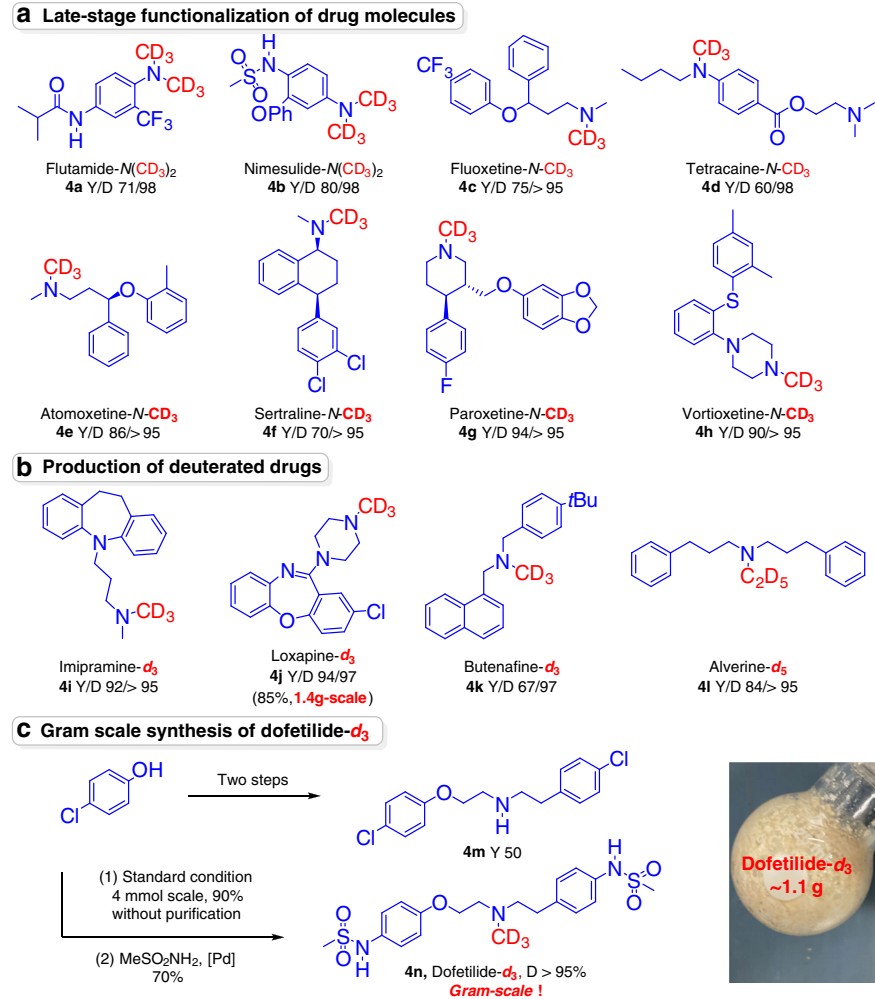

**Fig. 4 Late-stage functionalization and preparation of deuterated drugs. a** Late-stage functionalization of drug molecules. **b** Production of deuterated drugs. **c** Gram-scale synthesis of dofetilide-$d_3$.

and iminodibenzyl was achieved successfully. These heterocyclic skeletons are widespread in natural products, pharmaceuticals and key intermediates. Aliphatic amines were also found to be competent substrates, providing the desired products (**3da–3db**) in good yields. Finally, the strategy could be extended to the *N*-deuterated alkylation of amines by replacing $d_4$-methanol with other deuterated alkanols, such as $d_5$-ethanol for *N*-CD$_2$CD$_3$ incorporation (**3e**). This protocol exhibits highly efficient in production of deuterated *N*-alkyl chemicals with excellent D-incorporation, thus holding great potential application towards the synthesis of stable isotope-labeled compounds for synthetic mechanism study as well as LC/MS quantification[6,34,38].

**Sustainable synthesis of deuterated pharmaceuticals.** *N*-Me amine units are present in many of the 200 top-selling drugs produced in 2018 and are often required for their intended pharmacological functionality[53–58]. Deuterium substitution of the *N*-Me groups of these drugs is highly desired. We tested the protocol developed above for the synthesis of *N*-CD$_3$-based pharmaceuticals and bioactive molecules (Fig. 4). Here, the use of heterogeneous catalyst provides an ideal solution to avoiding poising these drugs from the molecular catalyst due to its easy removal. First, late-stage functionalization of drug molecules with primary and secondary amines was evaluated[59]. Di-*N*-

trideuteromethylation of flutamide and nimesulide was accomplished, providing the deuterated drug derivatives in good yields (71–80% yields) without affecting the amide and sulfamine functionalities. A variety of commercially available pharmaceuticals with secondary amine units, namely, fluoxetine, tetracaine, atomoxetine, sertraline, paroxetine and vortioxetine, smoothly underwent *N*-trideuteromethylation (**4c-4h**, 60–94% yields), reconfirming the universality of our strategy. More importantly, this mild and general process enables access to site-specifically labeled drugs in a single step. Deuterium-labeled analogs of butenafine could be obtained in 67% yield (**4k**). Trideuteromethylation of monomethylated desipramine and amoxapine gave imipramine-$d_3$ (**4i**, 92%) and loxapine-$d_3$ (**4j**, 94%), respectively. The use of C$_2$D$_5$OD/D$_2$O as an alkylation reagent successfully afforded alverine-$d_5$ (**4l**) in high yield (84%). In addition, synthesis of dofetilide-$d_3$ was achieved in four steps with 32% overall yields from low-cost and commercially available starting materials. Gram-scale syntheses of both loxapine-$d_3$ and dofetilide-$d_3$ with high yields were demonstrated, highlighting the practical utility of this protocol. Again, all D-labeled pharmaceuticals and their analogs gave excellent deuterium incorporation.

Isotope-labeled bioactive compounds are extensively used to study interactions with lipid membranes, proteins, nucleic acids, etc[60,61]. In particular, the controllable incorporation of partially

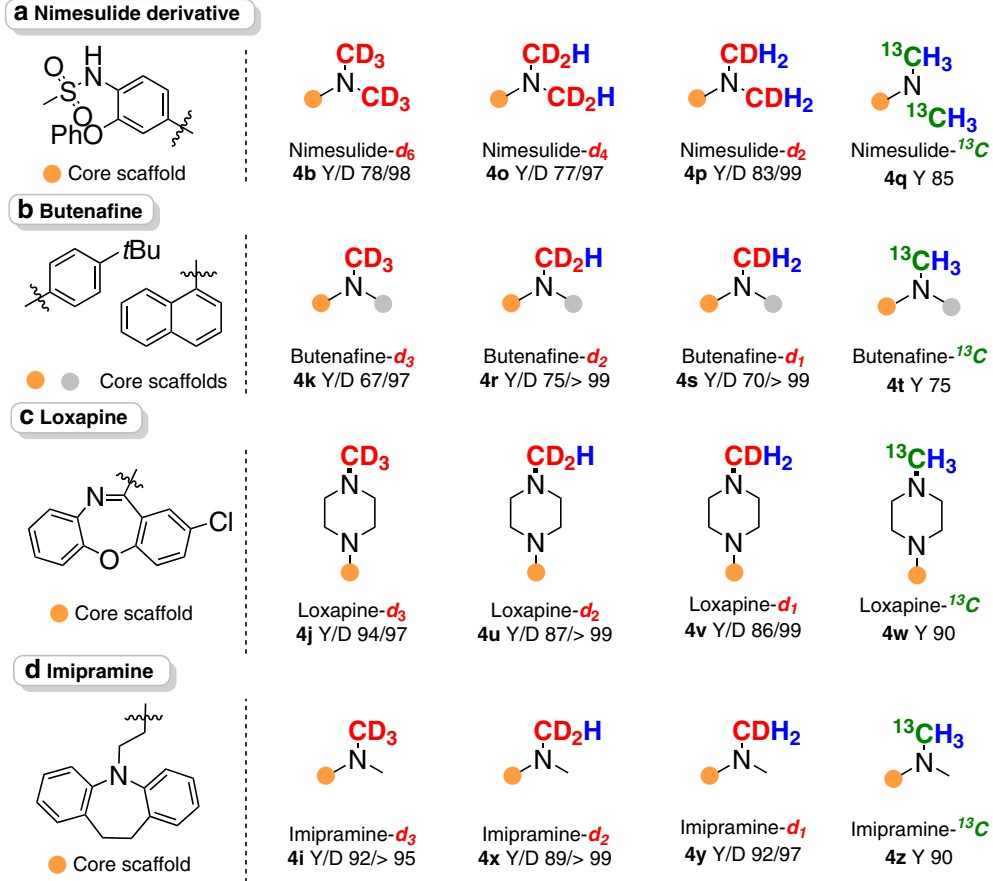

**Fig. 5 Schematic illustration of various isotopic drug preparations. a** Nimesulide derivative. **b** Butenafine. **c** Loxapine, **d** Imipramine.

deuterium-labeled *N*-methyl groups (CDH$_2$, CD$_2$H, or CD$_3$) slows drug metabolism to improve the pesticide effect[62]. However, their synthesis remains a great challenge. Our controllable deuterium-labeling strategy was successfully applied for the facile synthesis of *N*-CD$_3$, *N*-CD$_2$H and *N*-CDH$_2$ nimesulide derivatives (**4b**, **4o** and **4p**), butenafines-*d$_3$*, *d$_2$* and *d$_1$* (**4k**, **4r** and **4s**), loxapines-*d$_3$*, *d$_2$* and *d$_1$* (**4j**, **4u** and **4 v**) and imipramines-*d$_3$*, *d$_2$* and *d$_1$* (**4i**, **4x** and **4y**) with high yields and uniformly high D incorporation (>95%) (Fig. 5). In all these drugs, only the target *N*-alkyl units were specifically labeled with deuterium. $^{13}$C-labeled drugs are of significant importance in medical biology for tracking metabolites and quantitative analysis by mass spectrometry and $^{13}$C NMR spectroscopy[63]. This protocol can also be applied for the sustainable synthesis of $^{13}$C-labeled drugs by replacing methanol with $^{13}$CH$_3$OH. As expected, $^{13}$C-labeled nimesulide derivative (**4q**), butenafine (**4t**), loxapine (**4w**) and imipramine (**4z**) were readily obtained with comparable yields[64].

In summary, a powerful semiconductor photocatalytic system for the sustainable and scalable construction of deuterated pharmaceuticals and chemicals has been discovered. This strategy is characterized by high yields, excellent D incorporation in a single step, the use of low-cost and sustainable deuterated methylating reagents (isotopic water and methanol), excellent functional group tolerance including a range of pharmaceutically relevant functionalities, and mild conditions. Significantly, the unique controllable D-labeling protocol provides the ability to precisely control the number of deuterium atoms (i.e., *N*-CD$_3$, CD$_2$H and CDH$_2$) at the metabolic position of pharmaceuticals, which is critically important for deuterated drug discovery.

Finally, the present results reveal a new horizon of photosynthesis for direct pharmaceutical production.

## Methods

**Synthesis of CPCN and Pd/CPCN photocatalyst**. In a typical synthesis, melamine (3.0 g, Alfa Aesar) was ground with KBr (2.0 g, Alfa Aesar). Then, the resultant mixture was heated to 550 °C for 3 h in a tube furnace. After cooling to room temperature, the bright yellow-green product was washed with boiling deionized water several times and collected by filtration, followed by drying at 60 °C under vacuum. As-prepared sample is denoted as CPCN. Pd/CPCN photocatalyst was prepared by photodeposition process. In brief, as-synthesized CPCN (0.3 g) was dispersed in a mix solution with 80 mL deionized water and 20 mL glycol. After untrasonication treatment for 2 h, 84 μL of 1.0 M H$_2$PdCl$_4$ was added into the mixture, and then the mixture was treated under 300 W Xe lamp illumination for 1 h to reduce Pd$^{2+}$. The brownish slurry was centrifuged and washed with deionized water for three times. After dried in an oven at 70 °C overnight under vacuum condition, as-prepared sample denoted as Pd/CPCN were obtained.

**Photocatalytic deuterated *N*-methylation reaction**. Typically, 25 mg of Pd/ CPCN and 0.4 mmol of substrate and AlCl$_3$ (0.3 mmol) were dispersed in a mixture solution with Acetonitrile/D$_2$O/CD$_3$OD = 2 ml/1.5 ml/1.0 ml, and then sonicated for 1 min. The reaction mixture was then irradiated with a LED lamp (20 W, λ = 420 nm, Suncat instruments Co., Ltd., Beijing, China) for 4-24 h under Argon at 25°C by using a flow of cooling water during the reaction. After reaction, the mixture was centrifuged to remove photocatalyst. The supernatant was extracted by adding 5 mL of CH$_2$Cl$_2$. The reaction mixture was concentrated under reduced pressure and the residue was purified by column chromatography on silica gel to furnish the corresponding product. The isolated yield was calculated by dividing the amount of the obtained desired product. Deuterium incorporation were checked and calculated by NMR.

**Characterization equipment**. The crystal structure of catalyst was characterized by X-ray diffraction (XRD) (Ultima IV, Rigaku) at 40 kV and 40 mA (Cu Kα X-ray radiation source) with a scanning speed and step interval of 4° min$^{-1}$ and 0.01°, respectively. Transmission electron microscope (TEM) images were obtained using

a HT7700 TEM (Hitach). The solid diffuse reflectance spectra (DRS) were collected on a UV–Vis–NIR spectrophotometer (Cary 5000, Varian). NMR tests were conducted on Bruker AVANCE III NMR spectrometer (500 and 600 MHz). The high-performance mass spectrometry was conducted by a Q Exactive GC Orbitrap GC-MS/MS (Thermo Scientific).

## Data availability
All data are available from the authors upon reasonable request.

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

## Acknowledgements
This work was financially supported by the National Natural Science Foundation of China (21972094, 21902105), China Postdoctoral Science Foundation (2019M653004), Guangdong Special Support Program, Pengcheng Scholar program, Shenzhen Peacock Plan (KQJSCX20170727100802505 and KQTD2016053112042971), and Foundation for Distinguished Young Talents in Higher Education of Guangdong (2018KQNCX221). K.P.L. acknowledge NRF-CRP grant "Two-Dimensional Covalent Organic Framework: Synthesis and Applications". Grant number NRF-CRP16-2015-02, funded by National Research Foundation, Prime Minister's Office, Singapore.

## Author contributions
Z.Z. and C.Q. contributed equally to this work. C.S., Z.Z. and C.Q. designed this work. Y.X. and C.Q. synthesized and characterized the catalysts. Z.Z. and C.Q. optimized reaction conditions. Z.Z. accomplished the reactions and substrate scopes. Z.Z., C.Q., C.S., Q.H., J.T., and K.P.L. co-wrote the manuscript. C.S. supervised the research. All the authors discussed and commented on the manuscript.

## Competing interests
The authors declare no competing interests.
