## [Peer Review File · Nature Communications]

REVIEWER COMMENTS

Reviewer #1 (Remarks to the Author):

The authors describe the syntheses of deuterium labelled tertiary amines by photo-induced activation of alkanols and water followed by reaction with primary and secondary amines. This manuscript is without a doubt interesting and of good quality, however there are some aspects limiting the overall outcome.

The word "photosynthesis" is misleading in the title. I would use the word "photo-induced" as blue LEDs were used. This reaction can't be considered a mechanism like the biological process. I think the authors oversell this story. The same is true for page 3, line 8-17.

Page 2, line 13: the authors explain, that dealkylation is the major metabolic step for drug molecules. Unfortunately, they miss to give any reference. As this is their major selling point, I would at least expect 5-10 citations with a clear impact and explanation for deuterated drugs and their increased physiological half-life.

Page 2, line 23: The explanation for poor site selectivity in HIE reactions is just not true. Reference 20 or 21 describing very selective labelling reactions of tertiary amines in alpha and beta-position of aliphatic tertiary amines. These results especially the results described in ref. 20 should be discussed in more detail as it is a photo-induced method as well and the structures are very similar.

Furthermore, as the method is more likely comparable to alkylation reactions the authors should have discussed the advantages of their method in more detail. Even though deuterated methyl iodide is used in every isotope laboratory, the substitution is of interest as it is highly cancerogenic, volatile, and normally not delivered by airplane (long delivery times). Next to that, alkylation can lead to ammonium salts.

Fig 1, sorafenib: In this case the compound is a secondary amide. In all reported cases tertiary amines are shown. Please remove this compound from the scheme as it is misleading.

Based on Fig. 2 I would recommend adding an explanation on the mechanism of the reaction including the redox potential of the two cycles and to discuss the differences with ref. 20.

Is it possible to do a mono-alkylation by this protocol? I see only tertiary amines as products. In total, due to the missing discussion of already long-known alkylation (K_2CO_3 , MeI) or HIE reactions, the weird photosynthesis comparison and the missing discussions on the mechanism I can't recommend publication in Nature Communications.

Reviewer #2 (Remarks to the Author):

Shu et al. describe in this communication an interesting approach to synthesize isotopically labelled complex alkylamines. This approach is based on the oxidation of isotopically labelled alkanols to form in situ the corresponding aldehyde which reacts with an amine to form an imine reduced by [H]/[D] from water splitting.

The controlled access to d1 and d2 isotopologues (using H_2O/CD_3OD or D_2O/CH_3OH mixtures) of complex deuterated drugs containing N-alkylamine substructures described is elegant and very interesting but I think that the main synthetic problem to access to such deuterated drugs, namely the synthesis of the precursor, remains unsolved.

To my opinion the main problem of the current version of the manuscript is related to the selling points. The value of the target compounds (in terms of application) appears slightly oversold to me.

Indeed, the authors claim: "Deuterium substitution of N-alkyl groups affords high DKIEs in drug molecules and is highly effective in slowing drug metabolism and improving pesticide effects, especially in cytochrome P450 (CYP450)-catalyzed transformations." This is true for deuterium

substitution of methoxy groups (CYP mediated metabolism) or the substitution of some C-H bonds by C-D bonds in azines (MAO mediated metabolism). The effect of the deuteration of the pharmacokinetic of alkylamines is more marginal to the best of my knowledge (If I'm wrong I will be please to revise my opinion in the light of precise examples given by the authors).

Indeed, in the deuterated drug examples drawn in fig 1. only one molecule out of six (Morphine-d₃) only contains a deuterated alkylamine substructure to alter the drug's metabolism. In the other examples, methoxy, azine or amide substructures are also deuterated and these deuterations have a much more dramatic effect on the the rate of metabolization of the drug than the deuteration of the alkylamine substructure. If think the authors have to consider the actual number of examples of heavy drugs containing only deuterated alkylamine moieties and their benefit to alter drugs metabolism.

On the other hand (maybe because it looks less attractive), the authors did not mention the potential application of their method to synthesize stable isotopically labelled internal standards for LC/MS quantification. For me, the described method has a great potential of application in this context.

In the references list, some recent papers describing the deuteration of alkylamine substructures via hydrogen isotope exchange are missing while some research works, interesting to be mentioned in the context of tritium labelling of pharmaceuticals (17-18) of deuteration of aromatic ring (ref 19), are cited. This should also be improved.

All in all, I think this work is very interesting but I cannot recommend the publication of this paper in Nat. Comm. in the present form.

Reviewer #3 (Remarks to the Author):

The manuscript by Su and co-workers reports the development of Pd/CPCN (crystalline PCN)-catalyzed reaction of primary and secondary amines with CD₃OD and D₂O in the presence of 3/4 equivalent of AlCl₃. Under the irradiation by a LED lamp, a wide variety of alpha-deuterated amines can be produced in high efficiency. The method can be applied for the production of pharmaceuticals that possess N-CD₃, N-CD₂H, N-CDH₂ or N-¹³CH₃ group; functional group tolerance is therefore impressive (Figure 4a). This protocol may serve as a practical method to prepare the desirable d-labelled drugs (versus those that rely on the use of highly toxic D₃C-I).

Despite the strengths of the work as described above, this reviewer recommends that the authors make the following major revisions before it is considered for publication in Nature Communications.

- 1) The title is rather misleading. This method uses 3 weight% of a Pd-based catalyst, nearly a stoichiometric amount of AlCl₃ and is carried out in acetonitrile under the atmosphere of Ar. There is very limited relevance to "photosynthesis" and this term should not be used in the title.
- 2) In figure 3, the detailed reaction conditions should be given. The readers should not be directed to the SI (Table S1) to look for the conditions.
- 3) In the SI, it is noticeable that a majority of NMR spectra show that a significant amount of what seems to be grease is contained as an impurity. In addition, a number of samples (e.g., those for 4f, 4i, 4n, etc) are contaminated by a major impurities. Further purification should be carried out and appropriate spectra must be reported.

Point by point response to Reviewers

To Reviewer #1:

The authors describe the syntheses of deuterium labelled tertiary amines by photo-induced activation of alkanols and water followed by reaction with primary and secondary amines. This manuscript is without a doubt interesting and of good quality, however there are some aspects limiting the overall outcome.

Response: Thank you very much for the reviewer's positive comments and valuable suggestions. Herein, we have made our great efforts to address the review's major concerns, especially on discussion of the advantages of our method compared to known alkylation from CH_3I reactions, solid supports for the effect of the deuteration on the pharmacokinetics of alkylamines and more mechanistic details.

Question 1: *The word "photosynthesis" is misleading in the title. I would use the word "photo-induced" as blue LEDs were used. This reaction can't be considered a mechanism like the biological process. I think the authors oversell this story. The same is true for page 3, line 8-17.*

Response: Thank you for the reviewer's insightful comment. In natural photosynthesis, leaves can harness solar energy to split water to release oxygen and furnish [H] for sequential reduction process. This biological process inspired us to use semiconductor photocatalyst as an "artificial leaf" to synergistically decompose $\text{H}_2\text{O}/\text{D}_2\text{O}$ by photoexcited electrons to furnish [H]/[D]. Simultaneously, isotopic methanol could be oxidized to formaldehyde by photoexcited holes. Then, formaldehyde further react with organic substrates involving the photogenerated [H]/[D] to produce the target D-labeling *N*-alkyl Pharmaceuticals. Indeed, as the reviewer' said, we have realized our route is different from the well known biological photosynthesis process. Since the photoexcited eletrons and holes of semiconductor photocatalyst play significant roles in activating of water and methanol for *N*-alkylation of amines, we thus revised the word "photosynthesis" to "semiconductor photocatalysis" accordingly to avoid the unnecessary misleading.

Actions: We have corrected the title to “Semiconductor Photocatalysis to Engineering Deuterated N-alkyl Pharmaceuticals Enabled by Synergistic Activation of Water and Alkanols”, and revised the “Abstract” and “Introduction” accordingly to accurately express the ideas.

1. **The title**, “Photosynthesis to Engineering Deuterated N-alkyl Pharmaceuticals Enabled by Synergistic Activation of Water and Alkanols” **was revised to** “Semiconductor Photocatalysis to Engineering Deuterated N-alkyl Pharmaceuticals Enabled by Synergistic Activation of Water and Alkanols”
2. **In the abstract**, “Inspired by photosynthesis in which leaves harness solar energy to split water to furnish [H] (NADPH) for hydrogenation in the Calvin cycle, we herein rationally designed to synergistically derive D₂O-splitting to furnish [D] and alkanol-oxidation by photoexcited electron-hole pairs on a polymeric semiconductor, with an aim to achieve isotope-labeling N-alkylation of amines.” **was changed to** “With an aim to achieve controllable isotope-labeling N-alkylation of amines, we herein rationally designed to synergistically derive photocatalytic water-splitting to furnish [H] or [D] and isotope alkanol-oxidation by photoexcited electron-hole pairs on a polymeric semiconductor.”
3. **In the introduction, paragraph 3**, “Natural photosynthesis inspires intense efforts towards biomimetic systems for solar fuel production, while man-made system for pharmaceutical production from water and organics is a new blue print to be explored.³⁰ In natural photosynthesis, leaves harness solar energy to split water to release oxygen and furnish [H] (NADPH) for sequential reduction in the Calvin cycle.^{31, 32} Playing the same role as chloroplasts in leaves, a semiconductor photocatalyst can be used as an “artificial leaf” to decompose H₂O/D₂O to furnish [H]/[D]. Compared to natural photosynthesis, the advantage of using a man-made system lies in the synergistic activation of both water and organic substrates to generate reductive [H]/[D] and reactive organic species, which holds great potential for pharmaceutical production with rational design.” **was changed to** “In natural photosynthesis, leaves harness solar energy to split water to release oxygen

and furnish [H] (NADPH) for sequential reduction in the Calvin cycle.^{42, 43} With rational design, a semiconductor photocatalyst, which provide redox center on the surface under light irradiation can mimic as “artificial leaf” to similarly decompose H₂O/D₂O to furnish reductive [H]/[D] and simultaneously oxidize organics by the photoexcited electron-hole pairs. Synergistic utilization of those reductive [H]/[D] and reactive organic species holds great potential for production of deuterated chemicals and pharmaceutical, e.g. D-labeling N-alkyl Pharmaceuticals, from isotopic water and organics.”

Question 2: *Page 2, line 13: the authors explain, that dealkylation is the major metabolic step for drug molecules. Unfortunately, they miss to give any reference. As this is their major selling point, I would at least expect 5-10 citations with a clear impact and explanation for deuterated drugs and their increased physiological half-life.*

Response: Thank you very much for reviewer’s useful suggestions. We are sorry about the absence of relative references. Dealkylation of tertiary amines is indeed one of the important biochemical processes, which play major role in a number of *N*-alkyl drug metabolism, for example Clozapine [*Pharm. Res.* 2007, 24, 842], caffeine [*Biochem. Mol. Biol. Edu.*, 2006, 34 66; *Proc. Nati. Acad. Sci. USA*, 1989, 86, 7696], atrazine [*Dalton Trans.*, 2014, 43, 12175], benzphetamine, ephedrine, and methadone [*Rev. Physiol. Biochem. Pharmacol.*, 1995, 127, 138]. When substituting hydrogen for deuterium, for instance deuterium substitution in the methylene hydrogens, larger activation energy for the C-D bond may result in large primary (kinetic) hydrogen isotope effect (KIEs) for amine *N*-dealkylation reactions, and then lead to a reduction in the rate of oxidative *N*-demethylation. [*Science*, 1961, 134, 1078; *J. Label. Compd. Radiopharm.*, 2013, 56 428; *J. Am. Chem. Soc.* 1989, 111, 8646; *Chem. Commun.*, 2000, 393; *J. Biol. Chem.*, 1983, 258, 14445] Impressively, Elison etc. reported the deuteration of the *N*-methyl group of morphine not only caused reduction in potency, but also a distinct weakening of the binding to the enzyme active centers. [*Science*, 1961, 134, 1078] Therefore, deuterium substitution has been considered as a potential

means of slowing drug metabolism or redirecting sites of metabolism.

Actions: A series references were added to support our claim that “the *N*-dealkylation metabolized cytochrome P450 (CYP450) are commonly found in such *N*-alkyl drugs and other bioactive molecules” and “deuterium substitution of *N*-alkyl groups in *N*-alkyl drug molecules could contribute to slow down the N–C bond cleavage, and impacts their pharmacodynamic properties and improve pesticide effects.” The description is slightly revised to accurately express the ideas.

In the introduction, paragraph 1, “Among the myriad of commercial drugs, over 50% of the top sellers contain *N*-alkyl amine units,¹² and the key metabolic step for such drug molecules is often the *N*-dealkylation step (Fig. 1a). Deuterium substitution of *N*-alkyl groups affords high DKIEs in drug molecules and is highly effective in slowing drug metabolism and improving pesticide effects, especially in cytochrome P450 (CYP450)-catalyzed transformations.” **was changed to** “Among the myriad of commercial drugs, over 50% of the top sellers contain *N*-alkyl amine units¹³, and the *N*-dealkylation metabolized cytochrome P450 (CYP450) are commonly found in such *N*-alkyl drugs and other bioactive molecules.¹⁴⁻¹⁸ Thus, deuterium substitution of *N*-alkyl groups in *N*-alkyl drug molecules could contribute to slow down the N–C bond cleavage, and impacts their pharmacodynamic properties and improve pesticide effects.^{19-23,}”

Question 3: *Page 2, line 23: The explanation for poor site selectivity in HIE reactions is just not true. Reference 20 or 21 describing very selective labelling reactions of tertiary amines in alpha and beta-position of aliphatic tertiary amines. These results especially the results described in ref. 20 should be discussed in more detail as it is a photo-induced method as well and the structures are very similar.*

Response: Thank you for reviewer’s valuable suggestion and we have revised the description accordingly. Herein, a comparison of diverse strategies for construction of deuterated *N*-alkyl amines was added to discuss the difference of these results in more details.

Comparison of diverse strategies for synthesis of deuterated N-alkyl amines

a. Traditional synthetic method

b. Direct selective HIE reactions

c. This work

Scheme S2. Comparison of diverse strategies for synthesis of deuterated N-alkyl amines.

First, deuterated alkylation of amines with CD_3I is well-explored. As the reviewer said, these deuterated reagents are highly cancerogenic, volatile, and normally not delivered by airplane (long delivery times), thus hinders their practical applications. Next to that, alkylation with CD_3I can lead to ammonium salts. **Second**, transition metal-catalyzed HIE reactions at aromatic $C(sp^2)$ -H moieties are well established for deuterium and tritium labelling; directing groups are generally required. (Nature, 2016, 529, 195-199; Adv. Synth. Catal., 2014, 356, 3551-3562). The direct HIE at aliphatic $C(sp^3)$ -H moieties by transition metal-catalyst remains a challenge in the field. (Science, 2017, 358, 1182-1187; J. Am. Chem. Soc. 134, 12239-12244.) In ref 20, Beller group had demonstrated the transition metal HIE at both aliphatic α and β -amino $C(sp^3)$ -H bonds. **The key steps for this**

transformation include the generation of a reactive iminium cation or enamine intermediate followed by subsequent rehydrogenation. High temperature and complex noble metal catalyst is generally required. **Third**, in the *ref.* 21, MacMillan group reported a powerful photo-redox mediated HIE reaction which could efficiently and selectively install deuterium or tritium at α -amino sp^3 C–H bonds of the *N*-alkyl amine based drug molecules. **The key steps for this transformation include the generation of a reactive α -amino radical from *N*-alkyl amines catalyzed by a molecule photocatalyst followed by subsequent the hydrogen atom transfer (HAT) catalysis with D_2O or T_2O . Although these HIE methods are powerful and useful, they still can't be the substitutions to classic method with CD_3I . In this work, we have achieved a controllable isotope-labeling *N*-alkylation of amines with a combined alkylation reagents (isotopic water and alkanols). Compared to the photocatalytic HIE reaction (*ref* 21), the function and mechanism is very different. In term of the function, this work could be a good substitution to traditional deuterated methylation of amines with CD_3I as low cost isotopic water and methanol are utilized as the deuterated methylation reagent. More impressively, this work enables controllable installation of $-CH_3$, $-CDH_2$, $-CD_2H$, $N-CD_3$, and $-^{13}CH_3$ into pharmaceutical amines by facilely tuning isotopic water and methanol under mild conditions, which is very difficult to be achieved by these aforementioned methods. In term of the mechanism, the key step of this work is the synergistic utilization of electron-hole pairs generated on semiconductor PCN, in which photogenerated electrons are designed to reduce water to furnish reactive [H]/[D] species and photogenerated holes with appropriate oxidative ability were utilized to selectively oxidize isotopic methanol to produce aldehyde. Then, the reactive isotopic aldehyde and [H]/[D] species could be designed as the combined deuterated methylation reagent for deuterated alkylation of pharmaceutical amines under mild conditions (For the detailed comparison, please see the response to question 6).**

Actions: A comparison of diverse strategies for construction of deuterated *N*-alkyl amines as well as the detailed discussions was added in SI (please see scheme S2). The descriptions in the manuscript about the HIE reactions has been revised as follows: “Recently, transition metal-catalytic hydrogen isotope exchanges (HIE) has been emerged as an promising way to incorporate multi-deuterium or tritium atoms into α - or β -position of *N*-alkyl amines, generally albeit with harsh conditions or the use of complex noble metal catalysts (Scheme S2b).³¹⁻³⁷ In 2017, MacMillan group³⁸ reported a powerful photo-redox mediated HIE reaction which could efficiently and selectively install deuterium or tritium at α -amino sp^3 C-H bonds of the *N*-alkyl amine based drug molecules under mild conditions. In this protocol, the α -position of amines is oxidized by a molecular photocatalyst to yield α -amino radical, which was then trapped by the hydrogen atom transfer (HAT) catalysis to furnish α -deuterated or tritiated amine product. Multi deuterium atoms incorporation at all α -position of pharmaceutical amines (more than 4.0 deuteriums per molecule) with a wide range of D-incorporation ratio (from 1% to 91%) is generally occurred. Still, the development of a general and mild method for the substitution of the traditional deuterated alkylation from toxic deuterated reagents like CD_3I is in high demand to selectively functionalize pharmaceutical amines.³⁹⁻⁴³ Further, the precise control of deuterium atoms number at the α -position of *N*-alkyl drugs with high deuterium incorporation is currently remain unexplored, while it is particularly attractive for their potential use in mechanistic and metabolic studies.^{44, 45}

Question 4: *Furthermore, as the method is more likely comparable to alkylation reactions the authors should have discussed the advantages of their method in more detail. Even though deuterated methyl iodide is used in every isotope laboratory, the substitution is of interest as it is highly cancerogenic, volatile, and normally not delivered by airplane (long delivery times). Next to that, alkylation can lead to ammonium salts.*

Response: Thank you very much for reviewer’s helpful advice. We have added a detailed discussion on the advantages of our method compared to alkylation reactions

with CD₃I.

Actions: The discussion on the advantages of our method was added in Page 4 as follows: “Compared to traditional approaches from deuterated alkylation reagents, this photocatalytic strategy exhibits several advantages: a) the low-cost and sustainable isotopic water and alkanol is proposed as a new combined deuterated alkylation reagent; b) benefiting from this unique design, precise controlling the number of deuterium atoms (i.e., *N*-CD₃, CD₂H and CDH₂) at the metabolic position of *N*-Me drugs is enabled by simply tuning the isotopic water and methanol (Fig. 1c); c) excess deuterated methylation leading to ammonium salts could be effectively avoided; d) finally, this heterogeneous process exhibits high yields, broad reaction scope, excellent one-step D incorporation, and scalable production, thus paving the way towards deuterated drug studies and developments.”

Question 5: *Fig 1,:* In this case the compound is a secondary amide. In all reported cases tertiary amines are shown. Please remove this compound from the scheme as it is misleading.

Response: Thank you for reviewer’s helpful advice. We have removed the secondary amide compound from Figure 1 accordingly.

Actions: The secondary amide compound was removed from Figure 1 accordingly.

Question 6: *Based on Fig. 2 I would recommend adding an explanation on the mechanism of the reaction including the redox potential of the two cycles and to discuss the differences with ref. 20.*

Response: Thank you for reviewer’s valuable suggestions. To more accurately answer this question, we have revised Figure 1 and include very detailed mechanism as Figure 1b and compared with the mechanism of photocatalytic HIE as follows: **First, semiconductor photocatalysis is different with molecule photocatalysis. Light absorption by heterogeneous semiconductor photocatalysts generates surface redox centers as electron-hole pairs. As such, a semiconductor photocatalyst,**

upon photoexcitation, accomplishes two aligned redox transformations on the same particle surface, whereas a molecular photocatalyst, after electron transfer to one reaction partner, completes the overall redox process through a subsequent redox reaction of the oxidized or reduced catalyst. (Ghosh et al., *Science*, 2019, 365, 360–366) In the photocatalytic HIE report (*ref* 21, *Science*, 2017, 358, 1182-1187), photoexcitation of the iridium (III) photocatalyst generates the long lived triplet excited state Ir^{III} complex 2. The key step is that the excited species oxidize amine 3, which undergoes the deprotonation at the α -position to give α -amino radical 4. The radical 4 is then trapped by the hydrogen atom transfer (HAT) catalysis mediated the abstraction of deuterium from D₂O or T₂O to furnish α -deuterated or tritiated amine product and the electrophilic thiol radical 9. A second single-electron transfer was occurred between 5 and 9 to regenerate photocatalyst 1.

Fig. R1 The mechanism of photocatalytic HIE (from *Science*, 2017, 358, 1182-1187).

Differently, in this work, redox centers as electron-hole pairs are generated on semiconductor polymeric carbon nitride. Two aligned redox transformations named water-reduction and isotopic alkanol oxidation are accomplished on the same semiconductor surface. Amine substrates react with isotopic aldehyde from isotopic alkanol oxidation furnishes the iminium cation, which undergoes subsequent hydrogenation/deuteration by [H]/[D] to produce the corresponding

products.

b: Mechanistic proposal

Fig. R2 The mechanism of this work.

Actions: The detailed mechanism is newly added in the **Figure 1** and the discussion on the difference between this work with the photocatalytic HIE was added in the manuscript and SI.

Fig. 1 a: typical deuterated N-alkyl based drugs; b: designed reaction cycle of the controllable isotope-labeled N-methylation of amines by the synergistic utilization of electrons and holes on a semiconductor photocatalyst; c: this work, and the example of practical synthesis of isotope labeled loxapines.

1. **In the manuscript**, the detail explanation on the mechanism of the reaction was added in page 4 as follows: Upon visible-light irradiation, electron-hole pairs are generated on crystalline PCN. Photogenerated electrons are transferred to the anchored Pd nanoparticles and utilized to reduce water to furnish absorbed [H]/[D] species. Meanwhile, photogenerated holes with appropriate oxidative ability were designed to selectively oxidize isotopic alkanols, furnishing isotopic aldehydes for

aldehyde-amine condensation to produce imine intermediates. These imines were subsequently reduced by [H]/[D] from water splitting, producing corresponding N-alkyl chemicals and drugs (Fig. 1b).

2. A detail comparison on the mechanism between the photocatalytic HIE and our work is added in Page 4 of SI as follows: “Also, the mechanism in this work is very different with the reported photocatalytic HIE. In photocatalytic HIE, photoexcitation of the Ir^{III}-photocatalyst generates the long lived triplet excited state Ir^{III} complex, which oxidize amine at the α -position to give α -amino radical. Then the radical is trapped by the HAT catalysis to furnish α -deuterated or tritiated amine and the electrophilic thiol radical. A second single-electron transfer was occurred between Ir^{II} complex and electrophilic thiol radical to regenerate Ir^{III}-photocatalyst. In this work, redox centers as electron-hole pairs are generated on semiconductor polymeric carbon nitride. Two aligned redox transformations named water-reduction and isotopic alkanol oxidation are accomplished on the same semiconductor surface. Amine substrates react with isotopic aldehyde from isotopic alkanol oxidation furnishes the iminium cation, which undergoes subsequent hydrogenation/deuteration by [H]/[D] to produce the corresponding products.”

Question 7: *Is it possible to do a mono-alkylation by this protocol? I see only tertiary amines as products.*

Response: Thank you for reviewer’s comment. We have conducted the parallel experiments of the model reaction, and it is found that the mono-alkylation product could be obtained in about 22% yield within 5 h, and the product **3aa** was obtained in 89% yield after 9 h. Hence, mono-alkylation product is an intermediate of the overall alkylation reaction.

Fig. S4 Time-dependent yield of alkylation products

In the current stage, we could realize mono-alkylation by the following route: introducing a protecting group into amines such as (Bn); then the protected amine undergoes alkylation reaction followed by de-protection to furnish mono-alkylation product as shown as follows:

Fig. R3 Mono-alkylation of amines via protecting-group strategy.

In the future work, we are very interested to optimize the photocatalysts, the reaction conditions as well as the steric effect of the substrates to achieve efficient mono alkylation with modified protocol.

Actions: Supplementary experiments of the time-dependent alkylation products were conducted and results were illustrated in **Figure S4**. Please see the Supplementary data.

In total, due to the missing discussion of already long-known alkylation (K_2CO_3 , MeI) or HIE reactions, the weird photosynthesis comparison and the missing discussions on the mechanism I can't recommend publication in Nature Communications

Response: Thank you very much for reviewer's valuable and helpful comments, which are extremely useful for us to correct and improve our work. In the revised manuscript, the discussion of our method compared to already long-known alkylation (K_2CO_3 , MeI) and HIE reactions is provided; the improper using "photosynthesis" has been revised; the detailed mechanism as well as the discussion with the reported photocatalytic HIE has been provided. Thus, we wish this revised manuscript could address the reviewer's concerns.

To Reviewer #2:

Su et al. describe in this communication an interesting approach to synthesize isotopically labelled complex alkylamines. This approach is based on the oxidation of isotopically labelled alkanols to form in situ the corresponding aldehyde which reacts with an amine to form an imine reduced by [H]/[D] from water splitting. The controlled access to d1 and d2 isotopologues (using H₂O/CD₃OD or D₂O/CH₃OH mixtures) of complex deuterated drugs containing N alkylamine substructures described is elegant and very interesting but I think that the main synthetic problem to access to such deuterated drugs, namely the synthesis of the precursor, remains unsolved.

Response: Thank you very much for the reviewer positive comments and suggestions. Based on the reviewer's valuable and helpful suggestions, significant improvements have been added in the revised manuscript.

Question 1: *To my opinion the main problem of the current version of the manuscript is related to the selling points. The value of the target compounds (in terms of application) appears slightly oversold to me. Indeed, the authors claim: "Deuterium substitution of N-alkyl groups affords high DKIEs in drug molecules and is highly effective in slowing drug metabolism and improving pesticide effects, especially in cytochrome P450 (CYP450) catalyzed transformations." This is true for deuterium substitution of methoxy groups (CYP mediated metabolism) or the substitution of some C-H bonds by C-D bonds in azines (MAO mediated metabolism). The effect of the deuteration of the pharmacokinetic of alkylamines is more marginal to the best of my knowledge (If I'm wrong I will be please to revise my opinion in the light of precise examples given by the authors).*

Response: Thank you very much for the insightful suggestions. Indeed, as the reviewer mentioned, deuterium substitution of methoxy groups or the substitution of some C-H bonds by C-D bonds in azines are significant and well-explored to improve the pharmacokinetic properties. Although relatively less-explored, dealkylation of

alkylamines is still one of the important biochemical processes (*Chem. Res. Toxicol.*, 2018, 31, 68–80), which play major role in a number of drug metabolisms, for example Clozapine [*Pharm. Res.* 2007, 24, 842], caffeine [*Biochem. Mol. Biol. Edu.*, 2006, 34 66; *Proc. Nati. Acad. Sci. USA*, 1989, 86, 7696], atrazine [*Dalton Trans.*, 2014, 43, 12175], benzphetamine, ephedrine, and methadone [*Rev. Physiol. Biochem. Pharmacol.*, 1995, 127, 138]. If the rate of demethylation is dependent on the ease with which the C-H bond is oxidized, and if the biological actions are a function of such *N*-demethylation, then a change in the C-H bonding force would similarly affect both phenomena. When substituting hydrogen for deuterium, for instance deuterium substitution in the methylene hydrogens, larger activation energy for the C-D bond may result in large primary (kinetic) hydrogen isotope effect (KIEs) for amine *N*-dealkylation reactions, and then lead to a reduction in the rate of oxidative *N*-demethylation. [*Science*, 1961, 134, 1078; *J. Label. Compd. Radiopharm.*, 2013, 56 428; *J. Am. Chem. Soc.* 1989, 111, 8646; *Chem. Commun.*, 2000, 393; *J. Biol. Chem.*, 1983, 258, 14445] Impressively, Elison etc. reported the deuteration of the *N*-methyl group of morphine not only caused reduction in potency, but also a distinct weakening of the binding to the enzyme active centers. [*Science*, 1961, 134, 1078] In addition, in some cases, although the *N*-demethylation step is not the major metabolic step, deuterium substitution of *N*-alkyl groups in these drug molecules is still important to improve their pharmacokinetic properties such as the case of deuterated tramadol **(Please refer to the response of the next question)**. Herein, our present work clearly offers a new and mild strategy to construct a series of deuterium substituted *N*-alkylation precursors and pharmaceuticals with precisely-controlled deuterium atom numbers, and therefore benefits to further verify and discover more practical deuterated *N*-alkyl-drugs.

Actions: To more accurately express the ideas, the claim: “Deuterium substitution of *N*-alkyl groups affords high DKIEs in drug molecules and is highly effective in slowing drug metabolism and improving pesticide effects, especially in cytochrome P450 (CYP450) catalyzed transformations.” **was changed to** “Among the myriad of

commercial drugs, over 50% of the top sellers contain *N*-alkyl amine units,¹³ and the *N*-dealkylation metabolized cytochrome P450 (CYP450) are commonly found in such *N*-alkyl drugs and other bioactive molecules.¹⁴⁻¹⁹ Thus, deuterium substitution of *N*-alkyl groups in *N*-alkyl drug molecules could contribute to slow down the N-C bond cleavage, and impacts their pharmacodynamic properties and improve pesticide effects.²⁰⁻²⁴ Several references [*Chem. Res. Toxicol.* 2018, 31, 68–80; *Science*, 1961, 134, 1078; *J. Label. Compd. Radiopharm.*, 2013, 56, 428; *J. Am. Chem. Soc.* 1989, 111, 8646; *Chem. Commun.*, 2000, 393; *J. Biol. Chem.*, 1983, 258, 14445] are added to support the revised claim.

Question 2: *Indeed, in the deuterated drug examples drawn in fig 1. only one molecule out of six (Morphine-d3) only contains a deuterated alkylamine substructure to alter the drug's metabolism. In the other examples, methoxy, azine or amide substructures are also deuterated and these deuterations have a much more dramatic effect on the rate of metabolization of the drug than the deuteration of the alkylamine substructure. If think the authors have to consider the actual number of examples of heavy drugs containing only deuterated alkylamine moieties and their benefit to alter drugs metabolism.*

Response: Thank you for reviewer's helpful suggestions. We have revised the Fig. 1 on the basis of the reviewer's comments. More drugs only contains a deuterated alkylamine substructure was added in Fig. 1 and Fig. S1. In the other examples, where the methoxy, azine or amide substructures are also deuterated, the deuterated alkylamine substructure is still indispensable and benefits to alter drugs metabolism. For example, deuterated analogs of tramadol (**a**, **b**, **c**) have been evaluated in both in vitro and vivo systems. Only **c** had a significantly longer half-life ($t_{1/2}$) than tramadol which indicates that the deuterated alkylamine substructure benefit to alter drugs metabolism. [*Bioorg. Med. Chem. Lett.*, 2006, 16, 691]

Fig. R4 Deuterated analogs of tramadol.

Actions: More drugs only contain a deuterated alkylamine substructure was added in Fig. 1 and Fig. S1.

a: Deuterated *N*-alkyl based drugs

Fig. 1a typical deuterated *N*-alkyl based drugs;

Metabolism of *N*-alkyl based drugs: *N*-Dealkylations

Deuterated *N*-alkyl based drugs

Scheme S1. Deuterated *N*-alkyl containing drugs.

Question 3: *On the other hand (maybe because it looks less attractive), the authors did not mention the potential application of their method to synthesize stable isotopically labelled internal standards for LC/MS quantification. For me, the described method has a great potential of application in this context.*

Response: We thank the reviewer for bringing this important issue to our attention. The use of isotopically labeled internal standards is of particular advantage in the investigation of environmental, animal, and human samples in which matrix effects can interfere with the quantification of toxins. [*Chem. Eur. J.*, 2009, 15, 10397; *Angew. Chem. Int. Ed.*, 2014, 53, 230]. Herein, preparing stable isotopically labelled internal standards (SILSs) for LC-MS/MS investigations is of great significance. [*Angew. Chem. Int. Ed.*, 2007, 46, 7744; *Chem. Eur. J.*, 2009, 15, 10397; *Science* 2017, 358, 1182]. In the present work, deuterated compounds with high D-incorporation and controllable deuterium number in -methyl group could be easily synthesized under mild conditions. Thus, our method has great potential application to synthesize stable isotopically labelled internal standards for LC/MS quantification.

Actions: we have revised the discussions, and related references have been added in the manuscript as follows:

1. In introduction, page 1, “Isotope labeling plays vital roles in various fields in chemistry and the life sciences.¹⁻⁵” **was changed to** “Isotope labeling plays vital roles in various fields in synthetic chemistry, quantitative LC–MS/MS analysis and the life sciences.¹⁻⁶”
2. In page 7, paragraph 1, the following sentence was added: “This protocol exhibits highly efficient in production of deuterated *N*-alkyl chemicals with excellent D-incorporation, thus holding great potential application towards the synthesis of stable isotope labelled compounds for synthetic mechanism study as well as LC/MS quantification.^{6, 34, 38}”

Question 4: *In the references list, some recent papers describing the deuteration of alkylamine substructures via hydrogen isotope exchange are missing while some*

research works, interesting to be mentioned in the context of tritium labelling of pharmaceuticals (17-18) of deuteration of aromatic ring (ref 19), are cited. This should also be improved.

Response: Thank you for reviewer's helpful advice. References 17-19 have been removed. More recent references related to HIE reactions have been added in the manuscript, such as [*Chem. Lett.*, 2005, **34**, 192], [*Angew. Chem. Int. Ed.*, **2014**, 53, 230], [*J. Am. Chem. Soc.*, **2016**, 138, 13489] and [*ACS Catal.*, **2018**, 8, 10895].

Actions: We have revised the references accordingly.

All in all, I think this work is very interesting but I cannot recommend the publication of this paper in Nat. Comm. in the present form.

Response: Thank you very much for reviewer's valuable and helpful comments, which are extremely useful for us to correct and improve our current and future work. We hope the revised version is available to well-address these issues.

To Reviewer #3:

The manuscript by Su and co-workers reports the development of Pd/CPCN (crystalline PCN)-catalyzed reaction of primary and secondary amines with CD₃OD and D₂O in the presence of 3/4 equivalent of AlCl₃. Under the irradiation by a LED lamp, a wide variety of alpha-deuterated amines can be produced in high efficiency. The method can be applied for the production of pharmaceuticals that possess N-CD₃, N-CD₂H, N-CDH₂ or N-¹³CH₃ group; functional group tolerance is therefore impressive (Figure 4a). This protocol may serve as a practical method to prepare the desirable d-labelled drugs (versus those that rely on the use of highly toxic D₃C-I).

Response: We are very grateful to the reviewer for his/her comments and appreciation of our findings on development of green and general method for *N*-alkylation reactions which enable the construction of partially deuterium-labeled *N*-methyl groups (*N*-CDH₂ and *N*-CD₂H) under mild conditions. Based on the reviewer's valuable and helpful suggestions, significant improvements have been added in the revised manuscript.

Despite the strengths of the work as described above, this reviewer recommends that the authors make the following major revisions before it is considered for publication in Nature Communications.

Question 1: *The title is rather misleading. This method uses 3 weight% of a Pd-based catalyst, nearly a stoichiometric amount of AlCl₃ and is carried out in acetonitrile under the atmosphere of Ar. There is very limited relevance to "photosynthesis" and this term should not be used in the title.*

Response: Thank you for the insightful comment. The title as well as the description on photosynthesis in the manuscript has been revised according to the reviewer's suggestion.

Actions: We have corrected the title to "Semiconductor Photocatalysis to Engineering Deuterated *N*-alkyl Pharmaceuticals Enabled by Synergistic Activation of Water and

Alkanols”, and revised the “Abstract” and “Introduction” accordingly to accurately express the ideas.

4. **The title**, “Photosynthesis to Engineering Deuterated N-alkyl Pharmaceuticals Enabled by Synergistic Activation of Water and Alkanols” was revised to “Semiconductor Photocatalysis to Engineering Deuterated N-alkyl Pharmaceuticals Enabled by Synergistic Activation of Water and Alkanols”
5. **In the abstract**, “Inspired by photosynthesis in which leaves harness solar energy to split water to furnish [H] (NADPH) for hydrogenation in the Calvin cycle, we herein rationally designed to synergistically derive D₂O-splitting to furnish [D] and alkanol-oxidation by photoexcited electron-hole pairs on a polymeric semiconductor, with an aim to achieve isotope-labeling N-alkylation of amines.” **was changed to** “With an aim to achieve controllable isotope-labeling N-alkylation of amines, we herein rationally designed to synergistically derive photocatalytic water-splitting to furnish [H] or [D] and isotope alkanol-oxidation by photoexcited electron-hole pairs on a polymeric semiconductor.”
6. **In the introduction, paragraph 3**, “Natural photosynthesis inspires intense efforts towards biomimetic systems for solar fuel production, while man-made system for pharmaceutical production from water and organics is a new blue print to be explored.³⁰ In natural photosynthesis, leaves harness solar energy to split water to release oxygen and furnish [H] (NADPH) for sequential reduction in the Calvin cycle.^{31,32} Playing the same role as chloroplasts in leaves, a semiconductor photocatalyst can be used as an “artificial leaf” to decompose H₂O/D₂O to furnish [H]/[D]. Compared to natural photosynthesis, the advantage of using a man-made system lies in the synergistic activation of both water and organic substrates to generate reductive [H]/[D] and reactive organic species, which holds great potential for pharmaceutical production with rational design.” **was changed to** “In natural photosynthesis, leaves harness solar energy to split water to release oxygen and furnish [H] (NADPH) for sequential reduction in the Calvin cycle.^{42,43} With rational design, a semiconductor photocatalyst, which provide redox center on the surface under light irradiation can mimic as “artificial leaf” to similarly

decompose H₂O/D₂O to furnish reductive [H]/[D] and simultaneously oxidize organics by the photoexcited electron-hole pairs. Synergistic utilization of those reductive [H]/[D] and reactive organic species holds great potential for production of deuterated chemicals and pharmaceutical, e.g. D-labeling N-alkyl Pharmaceuticals, from isotopic water and organics.”

Question 2: *In figure 3, the detailed reaction conditions should be given. The readers should not be directed to the SI (Table S1) to look for the conditions.*

Response: Thank you very much for reviewer’s helpful suggestions. We have revised it accordingly.

Actions: The detailed reaction condition was added in the footnote of Figure 3 and the equation in Figure 3 has been revised.

Question 3: *In the SI, it is noticeable that a majority of NMR spectra show that a significant amount of what seems to be grease is contained as an impurity. In addition, a number of samples (e.g., those for 4f, 4i, 4n, etc) are contaminated by a major impurities. Further purification should be carried out and appropriate spectra must be reported.*

Response: Thank you very much for reviewer’s advice. We have rechecked all the NMR spectra and data of the products. The compounds **3ci**, **3cj**, **4f**, **4i**, **4n**, **4p**, **4r**, **4t** have been repurified and the appropriate spectra and data have been presented in the supporting information.

Action: The corresponding NMR spectra were added in SI.

REVIEWERS' COMMENTS:

Reviewer #1 (Remarks to the Author):

First, I must state that I am impressed by the authors response. Despite my general judgement that this method maybe borderline from the impact and general applicability I see the science and the serendipity to use new and modern approaches to improve old methods.

The answers and actions of the authors has improved the manuscript significantly. As last comment I would advise the authors to include the possibility to reduce N-CO₂R moieties with LiAlD₄ into their discussion of potential reaction alternatives. This would circumvent the formation of ammonium salts by alkylation. This method would be the strongest comparison. It would highly impress me if a compound like Osimertinib (figure 1) would be applied, which obviously can't be used with LiAlD₄.

At least the authors should include it in the discussion. With this change I recommend publication in Nature Communication.

Reviewer #3 (Remarks to the Author):

Su and co-workers have revised the manuscript extensively in response to the 3 reviewers' comments. This reviewer supports the publication of this intriguing work in Nature Communications as it stands.

Point by point response

Comments from review 1:

First, I must state that I am impressed by the authors response. Despite my general judgement that this method maybe borderline from the impact and general applicability, I see the science and the serendipity to use new and modern approaches to improve old methods. The answers and actions of the authors has improved the manuscript significantly. As last comment I would advise the authors to include the possibility to reduce N-CO₂R moieties with LiAlD₄ into their discussion of potential reaction alternatives. This would circumvent the formation of ammonium salts by alkylation. This method would be the strongest comparison. It would highly impress me if a compound like Osimertinib (figure 1) would be applied, which obviously can't be used with LiAlD₄. At least the authors should include it in the discussion. With this change I recommend publication in Nature Communication.

Response:

We are grateful for reviewer's approval of our work, and thank you very much for kindly reminding us another important approach to realize the N-CD₃ group. Typically, in 1961, Rapoport and coworkers carried out a study to probe the metabolism of morphine. To prepare deuterated morphine, normorphine was treated with excess ethyl chloroformate in the presence of potassium hydroxide to give *N*-dicarboethoxynormorphine, followed by LiAlD₄-induced reduction of the carbamate, affording the deuterated morphine in 73% yield (**Figure R1**). [*Science*, 1961, 134, 1078, *ref* 20 in the manuscript; *Deuterium*, 2016, Chapter 5 p99-p110] Although, reduction of *N*-CO₂R moieties with LiAlD₄ has good potential for the introduction of *N*-CD₃ group into pharmaceutical amines, the drawbacks of this approach are obvious, such as the need for introduction of extra functional group, the use of hazard and strong reduced reagent (LiAlH₄/LiAlD₄), poor functionalities tolerance e.g. amide, ester, cyano, ketone groups. In sharp contrast, our catalytic system by using alkanol and water as green combined reagent provides an efficient and mild method for the controllable installation of N-CH₃, -CDH₂, -CD₂H, -CD₃, and -¹³CH₃ groups into pharmaceutical amines with high efficiency, which clearly can't be accessed by LiAlD₄ strategy (**Figure 1**). More importantly, our method shows excellent functional group tolerance. A variety of functionalities that are sensitive to LiAlD₄ are well-tolerance in our method, such as the cyano group in compound **3ac**, ester group in compounds **3ae**, **3af**, **3ci**, **3cj**, **3ck**, ketone group in compound **3bi**, and amide group in compound **4b** etc. (**Figure R2**).

Figure R1. Synthesis of deuterium-labeled morphine by using LiAlD₄.

Figure R2. Products obtained by our method with sensitive functional groups to LiAlD₄.

Actions:

1. We have revised the **Supplementary Figure 2**, and the comparison of LiAlD₄ strategy have been added.
2. According to the reviewer's suggestion discussions on the applicability of LiAlD₄ strategy has been added into the manuscript.

In page 2-3: Reduction of *N*-CO₂R moieties with LiAlD₄ is another effective approach that has good potential for the introduction of *N*-CD₃ group without formation of ammonium salts.²⁰ However, introduction of extra functional group, use of hazard and strong reduced reagent, and functionality-tolerance limit its practical application (Supplementary Fig. 2a).

a. Traditional synthetic method

b. Direct selective HIE reactions

c. This work

- Deuterium substitution
- CD₃/CD₂H/CDH₂/CH₃

Supplementary Figure 2. Comparison of diverse strategies for synthesis of deuterated *N*-alkyl amines.